# EEG Sonification improves sleep staging performance in novice stagers

**Sam Chin**[1]☯, **Nathan W. Whitmore**[2]☯*, **Nathan Perry**[1], **Joe Paradiso**[1], **Pattie Maes**[2]

**1** Responsive Environments Group, MIT Media Lab, Cambridge, Massachusetts, United States of America, **2** Fluid Interfaces Group, MIT Media Lab, Cambridge, Massachusetts, United States of America

☯ These authors contributed equally to this work.

* nathanww@media.mit.edu

**Data availability statement:** Data and code used to perform all the analyses in this study are available at https://doi.org/10.5061/dryad.3bk3j9kz0.

## Abstract

Sleep staging is a critical tool used in research and clinical settings to evaluate and diagnose sleep conditions; however, sleep staging is labor intensive and may be challenging for inexperienced practitioners. We explored whether adding an auditory representation (sonification) of the EEG to a standard visual representation could improve sleep staging performance or reduce workload. This is the first study to investigate the effects of sonification on sleep staging performance. We performed a within-subjects study in which 40 participants completed an online sleep staging task with and without sonified EEG. EEG was sonified by minimal transformation in which the raw EEG signal was played as an audio signal. Contrary to our hypothesis, we found adding sonification did not result in improvements in accuracy, speed, or workload for the entire subject group. However, when we stratified participants by sleep staging experience, we found sonification improved accuracy for the least experienced participants. These findings suggest EEG sonification may be useful as a tool to enable novice sleep stagers to reach acceptable performance levels faster.

## 1 Introduction

During a night of sleep, humans transition through distinct physiological states, referred to as sleep stages. Sleep staging is a medical diagnostic procedure in which a technician classifies sleep into distinct sleep stages using a standardized set of rules applied to EEG, electromyogram, and electroculogram data [1].

Our study aimed to examine whether adding an audio representation of the EEG to a sleep staging interface could improve sleep staging performance. Prior research has investigated developing algorithms to convert EEG into sound [2–5], but has not yet addressed whether this technique can improve accuracy or reduce workload of sleep staging.

### 1.1 Manual sleep staging is challenging

Sleep staging is a frequently performed medical diagnostic procedure. In the United States, it is performed more than a million times per year in order to test for conditions such as sleep apnea [6]. These stages are then used to compute clinical sleep indices such as sleep onset

**Funding:** The author(s) received no specific funding for this work.

**Competing interests:** The authors have declared that no competing interests exist.

time, wake after sleep onset, and sleep fragmentation to diagnose specific sleep conditions such as apnea and periodic limb movement disorder [1,7].

Manual sleep staging is performed by examining electrical recordings of brain activity (EEG), eye movements (electrooculogram, EOG), and facial muscle tone (EMG) during the sleep period in 30-second epochs. Sleep scorers apply a standardized algorithm to classify each segment as Wake, REM sleep, or non-REM sleep stages 1-3. Sleep stages are identified using discrete visual visual features in these signals such as blinks, rapid eye movements, muscle twitches, and shifts in the frequency and waveform of the EEG. For example, wake is defined by the presence of 10-Hz oscillations in the EEG which are absent in sleep, REM sleep is defined by muscle paralysis and frequent rapid eye movements, and N1-N3 are associated with increasing amplitude of delta brain waves and the occurrence of specific waveforms like K-complexes and sleep spindles [1].

Manual sleep staging is highly labor intensive, and requires specialized training [8]. Furthermore, even highly trained sleep stagers show agreements only around 83% [8]. Thus, technology to improve sleep staging performance has the potential to improve both sleep staging accuracy and efficiency.

While machine learning algorithms have demonstrated good performance in sleep staging, [9] they are not used extensively in clinical practice due to concerns about their reliability and inability to provide justifications for their decisions. Regulatory agencies like the FDA have emphasized that medical AI algorithms should include a human-in-the-loop to verify decisions [10], thus manual sleep staging will likely continue to be necessary despite advances in AI sleep staging. It is therefore important to develop technologies which can facilitate faster and more accurate manual sleep staging.

## 1.2 EEG sonification for sleep staging

A potential approach for improving human judgment in sleep staging is EEG sonification, or presenting an auditory representation of the EEG. We take a multisensory approach and present the user with both an auditory and visual representation, since these can be integrated in the brain and improve perception of a stimulus [11,12].

There is significant prior art in sonifying bioelectric signals for analysis [13,14]. Some of the earliest work on neurons used an auditory interface to monitor activations [15]. Previous research has shown that sonification can reduce the time required to classify brain activity as epileptic or normal, suggesting that sonification can reduce the workload of EEG interpretation [2,3]. Research has also examined developing sleep-EEG sonification algorithms which balance aesthetic qualities with retaining important information in the EEG [4,5]. However, to date, no research has examined if adding sonification to an existing visual display system for sleep staging can improve accuracy or reduce workload of sleep staging.

## 1.3 How could sonification improve performance and reduce workload?

The brain combines information from sensory modalities (vision, audition, haptic, etc.) to form a unified percept of the world using multisensory integration. Multisensory integration maximizes reliability and minimizes variance of the integrated percept [16]. Having a multisensory stream of information allows each sense to function where it is most effective. For example, audition is more temporally precise, while vision is more spatially precise [17,18]. According to the Multiple Resource Model, a multisensory presentation may also reduce workload by moving parts of a complex visual task to an alternate sensory modality [19]. Sleep staging may be especially likely to benefit from multisensory representation because distinguishing sleep stages requires discriminating fine differences in the dominant frequency of

signals, which may benefit from the temporal and frequency precision of audition. In general by presenting both an auditory and visual signal, the brain can select the sensory modality with the most information.

For example, in a loud room, we use lip reading to augment our auditory perception of speech [20]. By combining both visual and auditory information, the brain creates a more reliable percept than either modality alone. This is also confirmed by experimental results which have demonstrated that sound can enhance visual perception [11,12,21].

Humans naturally learn to use multisensory integration implicitly, when useful information occurs in two or more sensory modalities. For example, through repeated exposure to the sight of objects falling and the accompanying sound, we develop an intuitive understanding of the physics of falling objects [22]. Adding sonification to a task has been demonstrated to assist in learning perceptual and motor skills [23,24] and interpretation of EEG data [2,3]. Therefore, mere exposure to sonified EEG (along with feedback) could potentially enable participants to learn to use sonic features to improve their performance sleep staging performance.

## 1.4 Study design

We tested whether sonifying EEG could improve accuracy, speed, and cognitive load using an online sleep staging task where participants were asked to stage sleep either with or without sonification. Specifically, we aimed to test the following hypotheses:

H1: Participants will perform sleep staging more accurately (as quantified by Cohen's kappa agreement with gold standard scorers) when they view an EEG with an accompanying sonified representation compared to viewing the EEG alone.

H2: Participants will stage faster when they view EEG and also hear a sonified representation than when they view the EEG without sonification.

H3: Participants will report lower mental demand, effort and frustration as measured by the NASA Task Load Index [25] when staging with sonification as compared to visual staging alone.

H4: Benefits will be selective for sleep epochs which contain alpha waves or sleep spindles.

## 2 Materials and methods

We preregistered our procedure prior to analysis at https://osf.io/2xz5e. Our analysis followed the procedures laid out in the preregistration with one exception. We chose not to analyze the number of transitions between epochs since fewer than 5 participants had transitions. The research was exempted from review by the MIT IRB (protocol #E-4933 ) and written consent was obtained online.

## 2.1 Participants and screening procedure

We recruited participants from March 8 2024 to March 18 2024 via advertisements distributed to sleep researchers on Twitter. Participants were eligible if they were at least 18 years of age, and reported some experience with AASM sleep staging. After observing a large number of probable bots attempting to complete the survey we also added an additional screening question requiring participants to correctly identify an obvious epoch of Wake to proceed; this question was added after the 386th participant. 536 participants completed screening and proceeded to the practice test.

To exclude inattentive participants and bots, we excluded participants with implausibly short reaction times in the practice block. Participants were not allowed to proceed to

the test blocks if their median time to stage an epoch during the practice block was less than 2 s. Any participants who failed this criterion were excluded from the data. The criterion was selected based on the reaction time distribution of 11 manually identified likely bots ($\mu$ of medians 0.59 s, $\sigma = 0.27$ s) vs 3 known valid sleep stagers ($\mu$ of medians 5.5 s, $\sigma = 2.1$ s).

There were 623 attempts to complete the practice block; of these, 46 met the reaction time criterion and proceeded to the test blocks. 40 of these participants completed all of the test phases and were included in analysis. The included participants consisted of 24 males, 15 females, and 1 non-binary person with a mean age of 24 ± 3.11 years. Reported sleep staging experience is shown in Table 1.

## 2.2 Stimulus generation procedure

We used data from healthy participants in the ISRUC-SLEEP database [26] downloaded on 27/02/2024. We included only sleep epochs which were staged as the same sleep stage by both ISRUC-SLEEP stagers. For more details on the sleep dataset, see [26].

We created the visual and auditory representations of the data using a modified version of Visbrain Sleep [27], an open-source sleep staging tool. An example of the visual display is shown in Fig 1. The data were displayed in an AASM standard format [1] for sleep staging with 5 channels displayed: 3 EEG channels (F3, C3, O1), 1 electroculogram channel (left outer canthus), and 1 chin EMG channel. All data were referenced to the right ear. All channels were scaled to ± 100 $\mu$V. Participants viewed one 30 second-long epoch of sleep at a time.

To generate the audio data we performed a minimal transformation where we converted 30 seconds of visually displayed data from the O1 channel into a 1.36 second sound clip. We used this approach rather than a more complex transformation to avoid destroying any information and to maximize participants' ability to learn. We chose a simple transformation over a musical or mapping-based approach because these risk increasing the salience of features irrelevant to sleep staging. A simple transformation allows the brain's auditory system to naturally identify the relevant properties of the sound. This approach enables the formation of perceptual boundaries based on the sound's intrinsic characteristics [28]. The original dataset was 30 s epochs measured at 200 Hz [29]. This was downsampled to 100 Hz in Visbrain. In order to produce the auditory stimulus, we sped up the visual data by approximately 95% for a final audio length of 1.36 s. We also performed scaling and padding of the audio to reduce clipping and popping during playback. We observed that this method produced qualitatively distinct sounds for each sleep stage. The resulting audio samples were low-frequency (20–1000 Hz) and were mixed to -12 dB (approximately standard loudness for musical audio tracks). Examples of the sounds (along with accompanying visual representation) can be found at https://github.com/nathanww/SoundspindleExperiment/tree/main/data/.

Table 1. Count of participants by sleep staging experience level.

| Number of Polysomnograms Staged in Past Year | Count |
| --- | --- |
| 1-10 | 14 |
| 11-20 | 8 |
| 21-40 | 10 |
| 41-80 | 5 |
| More than 80 | 3 |

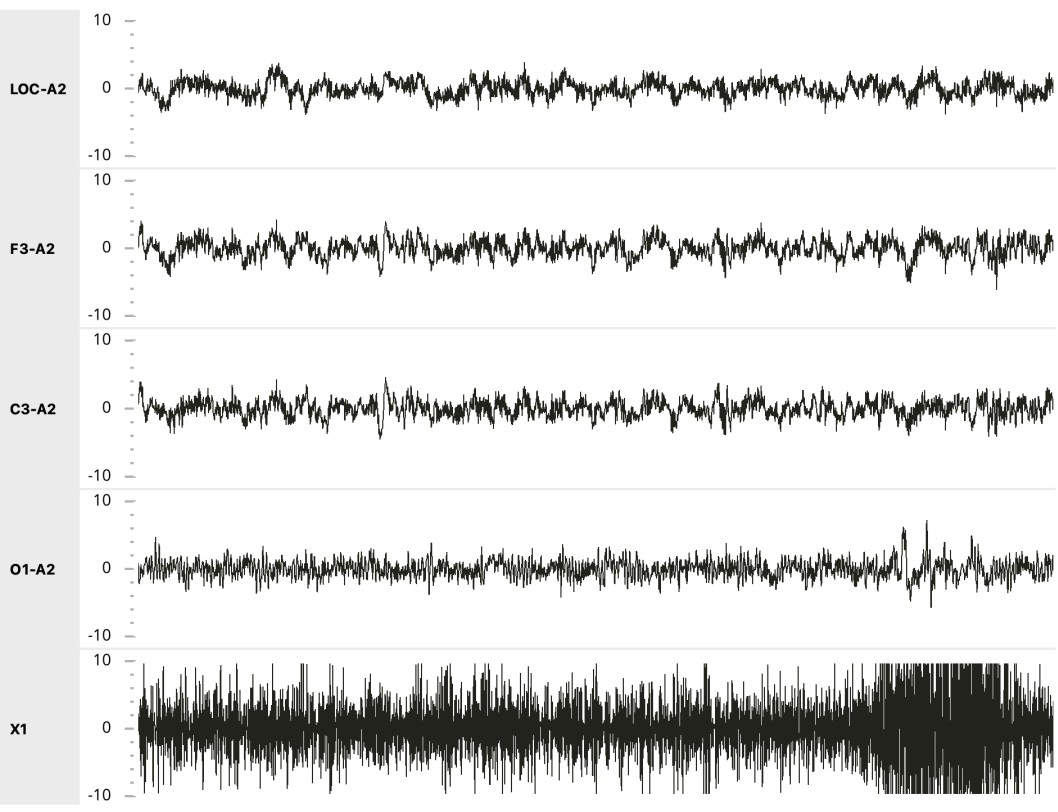

**Fig 1. An example of the online sleep staging task interface used by participants.** The O1-A2 channel was sonified using our stimulus generation procedure.

### 2.3 Experimental procedure

On beginning the experiment, participants first completed a questionnaire on their age, sex, and sleep staging experience. To measure experience, participants were asked how many polysomnograms they had scored in the past year with five possible choices: 1-10, 11-20, 21-40, 41-80, or more than 80 (Table 1). We choose to measure a "snapshot" of scoring in the past year (rather than lifetime scoring) as we predicted participants would be more accurate at estimating scoring over past year.

Participants then proceeded to the online sleep staging task summarized in Fig 2. The task began with an orientation to the display format (Fig 1), after which participants began the practice block of sleep staging. In this block, participants viewed and attempted to stage 20 30-second epochs of sleep. Participants were also able to toggle between the epoch to be staged, and the 30 seconds before and after this epoch as many times as desired before issuing a sleep stage. Sound corresponding to the O1 channel of the EEG was played immediately upon viewing the sleep epoch and any time the participant moved forward or backward in time.

After the participant issued a sleep stage, the sonification of the O1 channel was played again and the participant received feedback on the correct sleep stage. The participant could then proceed to the next epoch of sleep. Epochs were presented in random order until the participant issued the correct response; the epoch was then dropped from the rotation. Once all epochs were correctly identified, the block ended.

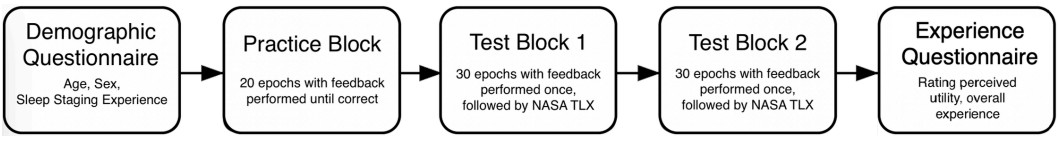

**Fig 2. The experimental procedure.**

Participants then proceeded to the first and second test blocks, which presented information identically to the practice block. Each included 30 epochs of sleep, and each epoch was shown only once, regardless of whether it was correctly staged. Participants were randomized and counterbalanced such that half received sonification on the first test block (but not the second) and half received sonification on the second test block (but not the first).

At the end of each test block, participants also performed the NASA Task Load Index [25] to assess cognitive load while performing the task. Following completion of the last block, we asked participants for their overall impressions on the experiment.

## 2.4 Statistical analysis

For all statistical analyses, we compared performance in the no-sound block to performance in the sound block using a paired test. We used a Wilcoxon signed-rank test for all comparisons except for the comparison of task load index scales, where the differences between blocks met normality assumptions for using a paired t-test. For all analyses, we defined the correct sleep stage for each epoch as the stage given by the ISRUC stagers.

We measured sleep staging accuracy using Cohen's kappa to measure agreement between the participant's stages and the correct stages in each block. Cohen's kappa provides an agreement score from -1 (complete disagreement) to 1 (perfect agreement) corrected for the agreement expected from random chance [30]. Importantly, kappa compensates for the highly imbalanced classes found in sleep staging which can distort simpler measurements of percentage correct.

To determine whether participants' sleep staging experience mediated the effect of sound, we split participants into 5 categories based on the number of polysomnograms they reported staging in the prior year. We then performed a paired test comparing kappa for the sound vs no-sound block for each experience group.

When analyzing reaction times, we only included trials where the participant gave the correct sleep stage. We excluded trials with outlier reaction times (RTs) more than two sigma above or below the average reaction time consistent with standard practices [31].

We hypothesized that sonification might selectively improve the ability to identify Wake and N2, which were associated with especially distinctive sounds. To determine if sonification improved participants' ability to identify specific sleep stages, we defined the true stage for each epoch as the stage given by the ISRUC stagers. We then measured the percent of "true stage X" that was identified as that stage by participants. For example, if participants recognized 5 of the 10 Wake epochs present in a block, their accuracy for Wake would be 50%. For this analysis, we quantified participant performance using percent correct rather than Cohen's kappa because kappa is undefined when the correct sleep stage is the same for all epochs.

We also computed how sonification affected the time required to identify stage Wake and stage N2 specifically (Fig 2). For these analyses, we only included trials where the RT was within 2 sigma of the mean and the sleep stage was correctly identified; 3 participants failed to

correctly identify any instances of stage Wake in one or more blocks and were excluded from this analysis.

## 3 Results

### 3.1 Sonification improves staging accuracy for participants with least experience

We found no significant difference between sleep staging performance in the sound and no-sound block [Wilcoxon signed-rank test, p = 0.21], and therefore no support for H1. Similarly we did not find support for H2, as the time to complete sleep staging did not differ significantly between the sound and no-sound blocks for all epochs staged [Wilcoxon signed-rank test, p = 0.65] or epochs staged correctly [Wilcoxon signed-rank test, p = 0.83].

However, our exploratory analysis revealed that sound improved accuracy measured by kappa only for the group of participants who reported having staged 1–10 nights of sleep previously; for this subgroup kappa was significantly higher with sound [Wilcoxon signed-rank test, p = 0.01] (Fig 3). No other group of participants showed a significant effect of sound.

### 3.2 Sonification did not significantly alter task load

To test our prediction that sonification would reduce the demand of sleep staging [H3], we measured whether NASA Task Load Index of mental load, effort, and frustration differed between the sound block and the no-sound block. As the differences were normally distributed, we used a paired t test. We did not observe any significant differences between the sound and no-sound conditions on any of the three axes (Fig 4). We also did not observe significant effects on the TLX for the subgroup of least experienced users (who reported scoring 1-10 nights of sleep).

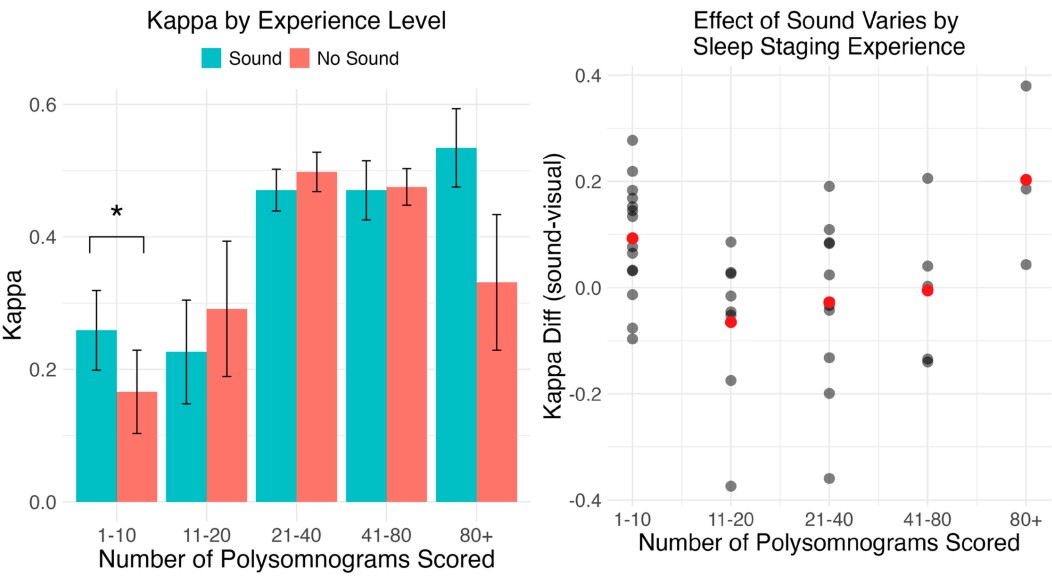

**Fig 3. (Left) Sleep staging performance (Cohen's kappa) of participants in the sound and no-sound blocks by experience level.** (Right) Difference in sleep staging performance between the sound and no-sound blocks. The least experienced participants showed a significant improvement in kappa with sound (indicated by *). Higher kappa indicates better staging performance.

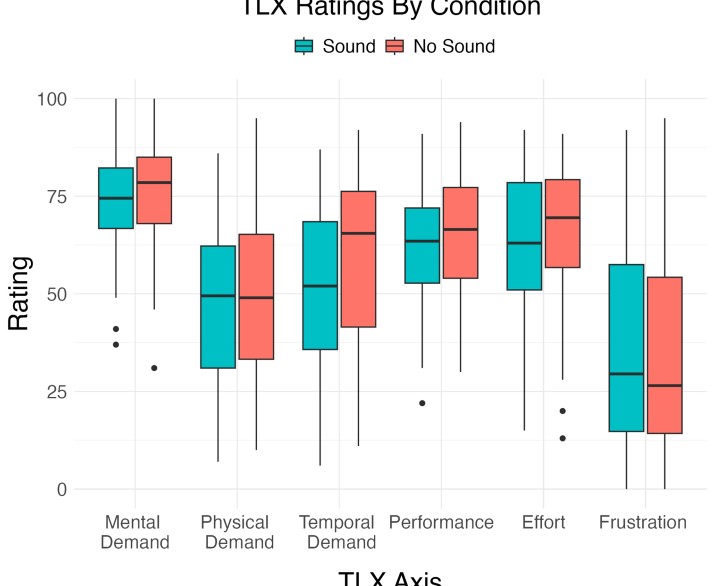

**Fig 4. The effect of sonification on NASA TLX scores was not statistically significant. Mental Demand** [t(39) = 1.052, p = 0.30], **Physical Demand** [t(39) = 0.524, p = 0.60], **Temporal Demand** [t(39) = 1.622, p = 0.11], **Performance** [t(39) = 0.826, p = 0.41], **Effort** [t(39) = 0.434, p = 0.67], **Frustration** [t(39) = -0.510, p = 0.61].

### 3.3 Effects of sonification did not depend on sleep stage

We predicted [H4] that sonification might selectively improve the recognition of Wake and N2 as those sleep stages contain features that are particularly identifiable in sound. However, we found that neither accuracy in identifying Wake nor accuracy in identifying N2 was improved by sonification (Fig 5), [Wilcoxon signed-rank test | Wake: p = 0.57, N2: p = 0.28]. Therefore we did not find support for this hypothesis.

### 3.4 Sleep staging experience predicts test performance

We observed a significant correlation between the number of polysomnograms staged in the past year and the overall kappa across both test blocks [r(38) = 0.49, p = 0.001].

### 3.5 Participants found sonification useful

After completing the experiment, 30/40 participants reported some benefit from the sound in staging sleep. 26 participants reported that the sound was "Useful, a lot" for staging, while 4 participants reported it was "Useful, a little", 4 indicated that they were unsure, and 6 reported that it was not useful. 39/40 participants indicated that they would use a sleep staging platform that incorporated sonification.

## 4 Discussion

In this study, we found that adding sonification to an EEG sleep staging task improved sleep staging accuracy only for the group of participants with the least experience. Contrary to predictions, we did not find that sonification reduced the time to complete staging or the mental load of the staging task; though participants reported finding the sounds useful.

## Percent Correct Response by Sleep Stage

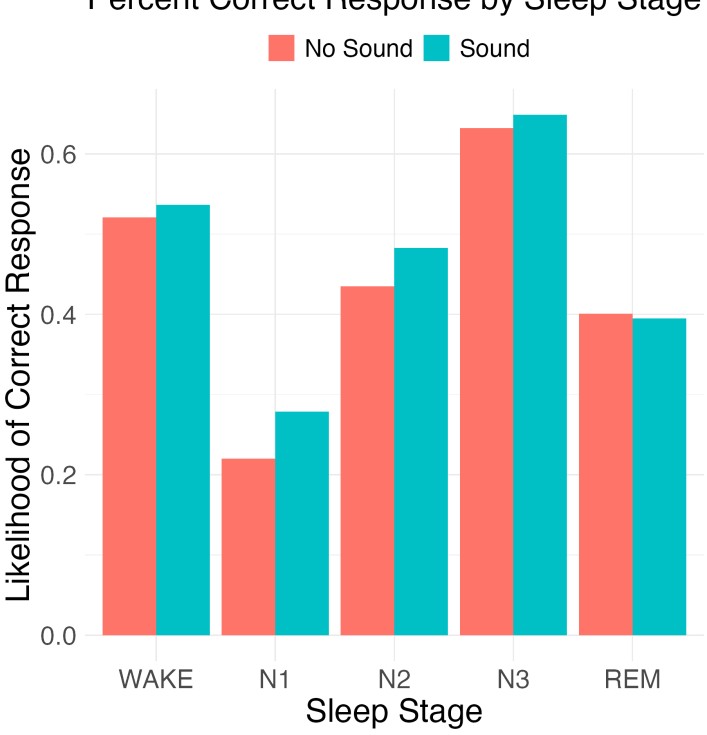

**Fig 5. Effect of sonification on accuracy for each sleep stage.** The correct sleep stage for an epoch was defined by the ISRUC sleep stagers. Accuracy by sleep stage was calculated as the number of epochs in the sleep stage that were correctly identified by the participant. We did not observe significant differences in the effect of sonification by sleep stage.

Why did we find sonification improved performance in novices, but not in experienced sleep stagers? One hypothesis is that there may be a ceiling effect where experienced sleep stagers can extract all the information in the signal from its visual representation; thus sound does not add any additional information. An alternative hypothesis is that experienced sleep stagers may experience a blocking effect where well-learned associations between visual features and sleep stages prevent effective learning of the relationship between sound features and sleep stage. Blocking is a well-documented effect in learning where once people reach good performance on a task they fail to incorporate other useful information because of a lack of prediction errors [32,33]).

Importantly, prior work on the effects of sonification on task performance [2,3,21,24] have typically used tasks that are at least somewhat novel to participants, where a blocking effect would not typically be observed. In contrast, the majority of our participants had extensive experience with visual sleep staging. This difference in prior experience (along with blocking or ceiling effects) why we did not observe the large sonification benefits observed in prior studies.

While we did not find overall significant effects of sonification, participants did frequently report that sound was helpful for sleep staging, and we observed non-significant improvements in the time to complete sleep staging. It is possible that more experienced participants would also show benefits from sonification with a larger sample size or with a longer period of learning.

Future studies could further explore the benefits of sonification for novice sleep stagers by comparing the rates at which naive participants improve in sleep staging when trained with visual plus sonified information compared to participants training with visual information alone. Because blocking effects should not occur when there is no pre-existing knowledge, novice participants trained with sonification may learn faster and attain higher levels of performance than those trained with visual information alone. Future studies could also explore the effects of alternative sound transformations or whether an increased number of learning trials improves the effects of sonification.

As a compensated online study, there is a risk of participants engaging in deception to collect rewards; risks here include both automated and semi-automated bots and human participants who do not perform the task according to instructions [34]. We employed multiple measures to detect and exclude these response types, including reCAPTCHA, excluding participants who failed to identify an obvious epoch of Wake, and excluding participants who exhibited implausibly fast responses during the practice block. We also demonstrated that the reported level of sleep staging experience correlated with performance, suggesting that the bulk of our participants correctly reported their experience level and attempted to perform the sleep staging task. Nonetheless, it is possible that the recorded answers contain some non-genuine responses.

A unique feature of this study compared to prior work on sonification of bioelectric signals is that we examined whether sonification combined with standard techniques could augment performance. In contrast, previous studies have examined the optimal techniques to create aesthetically pleasing sounds [4,5] or whether sonification could replace visual interpretation [2,3]. We found that sonification improved accuracy in novice sleep stagers and was generally found to be helpful by most users. Our observation that sonification can increase sleep staging accuracy in inexperienced sleep stagers suggests that it may be particularly valuable for non-specialists with limited sleep experience (such as internists or students). Future research should also explore whether augmenting sleep staging training programs with sonification can speed learning and improve performance.

## Author contributions

**Conceptualization:** Sam Chin, Nathan W. Whitmore.

**Data curation:** Sam Chin.

**Formal analysis:** Sam Chin, Nathan W. Whitmore.

**Funding acquisition:** Joe Paradiso, Pattie Maes.

**Investigation:** Sam Chin, Nathan W. Whitmore.

**Methodology:** Sam Chin, Nathan W. Whitmore.

**Project administration:** Sam Chin, Nathan W. Whitmore.

**Software:** Sam Chin, Nathan W. Whitmore, Nathan Perry.

**Supervision:** Joe Paradiso, Pattie Maes.

**Visualization:** Sam Chin, Nathan W. Whitmore.

**Writing – original draft:** Sam Chin, Nathan W. Whitmore.

**Writing – review & editing:** Sam Chin, Nathan W. Whitmore, Nathan Perry, Joe Paradiso, Pattie Maes.

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
