## [Decision Letter · Decision Letter 0]

3 Jun 2025

PONE-D-25-11016EEG Sonification Improves Sleep Staging Performance in Novice StagersPLOS ONE

Dear Dr. Whitmore,

Thank you for submitting your manuscript to PLOS ONE. After careful consideration, we feel that it has merit but does not fully meet PLOS ONE’s publication criteria as it currently stands. Therefore, we invite you to submit a revised version of the manuscript that addresses the points raised during the review process.

We look forward to receiving your revised manuscript.

Kind regards,

Hesam Ramezanzade, Ph.D

Academic Editor

PLOS ONE

**Journal Requirements:**

1. When submitting your revision, we need you to address these additional requirements. Please ensure that your manuscript meets PLOS ONE's style requirements, including those for file naming. The PLOS ONE style templates can be found at https://journals.plos.org/plosone/s/file?id=wjVg/PLOSOne_formatting_sample_main_body.pdf and https://journals.plos.org/plosone/s/file?id=ba62/PLOSOne_formatting_sample_title_authors_affiliations.pdf 2. Please update your submission to use the PLOS LaTeX template. The template and more information on our requirements for LaTeX submissions can be found at http://journals.plos.org/plosone/s/latex. 3. Thank you for uploading your study's underlying data set. Unfortunately, the repository you have noted in your Data Availability statement does not qualify as an acceptable data repository according to PLOS's standards. At this time, please upload the minimal data set necessary to replicate your study's findings to a stable, public repository (such as figshare or Dryad) and provide us with the relevant URLs, DOIs, or accession numbers that may be used to access these data. For a list of recommended repositories and additional information on PLOS standards for data deposition, please see https://journals.plos.org/plosone/s/recommended-repositories.

Reviewers' comments:

Reviewer's Responses to Questions

**Comments to the Author**

1. Is the manuscript technically sound, and do the data support the conclusions?

Reviewer #1: Yes

Reviewer #2: Partly

2. Has the statistical analysis been performed appropriately and rigorously? 

Reviewer #1: Yes

Reviewer #2: Yes

3. Have the authors made all data underlying the findings in their manuscript fully available?

Reviewer #1: No

Reviewer #2: Yes

4. Is the manuscript presented in an intelligible fashion and written in standard English?

Reviewer #1: Yes

Reviewer #2: Yes

5. Review Comments to the Author

**Reviewer #1: **This study is an interesting study on the effect of auditory intervention on sleep staging that has not been investigated so far. Most studies in the field of sanitization have used this type of intervention to develop (cognitive-motor) performance and have shown positive results.

1- The abstract is well-written; however, it would be beneficial to include a brief one-line reference to the feature that was presented to participants as an auditory pattern (sonification).

2- The introduction is well-structured. However, it would be beneficial to examine the effects of sonification on other functions, including motor performance, before discussing research related to sleep staging performance. Numerous studies have been conducted in this regard, some of which I have referenced. For esample:

Ramezanzade, H. (2019). Adding Acoustical to visual movement patterns to retest whether imitation is goal- or pattern- directed. Perceptual and Motor Skills, 127(1), https://doi.org/10.1177/0031512519870418.

Parimi F, Abdoli B, Ramezanzade H, Aghdaei M (2024) The Effect of Internal and external imagery on learning badminton long serve skill: The role of visual and audiovisual imagery. PLoS ONE 19(9): e0309473. https://doi.org/10.1371/journal.pone.0309473.

Effenberg AO, Fehse U, Schmitz G, Krueger B, Mechling H. Movement sonification: effects on motor learning beyond rhythmic adjustments. Frontiers in neuroscience. 2016;10:219. pmid:27303255.

3- You can reference the following source for the statement "audition is more temporally precise, while vision is more spatially precise".

Ramezanzade, H., Abdoli, B., Farsi, A., & Sanjari, M. A. (2017). The effect of sonification of visual modeling on relative timing: The role in motor learning. Acta Kinsiologica, 11, Supp. 2, 17-27.

4- You can also reference the following source for the statement “This is also confirmed by experimental results which have demonstrated that sound can enhance visual perception”.

Ramezanzade, H., Abdoli, B., Farsi, A., & Sanjari, M. A. (2017). The effect of sonification modelling on perception and accuracy of performing jump shot basketball. International Journal of Sports Studies, 4(11), 1388-1392.

Effenberg AO. Movement sonification: Effects on perception and action. IEEE multimedia. 2005;12(2):53–9.

5- Please specify precisely why the O1, F1, and C3 regions were selected for generating the auditory pattern.

6- The methodology is well-written, but it should include more details regarding the physical characteristics of the sound (frequency, volume, pitch, timbre, etc.)

7- It would be better to describe sleep staging precisely and in detail in both the introduction and methodology sections, while also specifying the characteristics of each stage.

8- The results section is well-reported. The discussion is also good, but it would be better to provide more speculation regarding the reasons behind the obtained results. It would be beneficial to study recent research related to sleep and its influencing variables, and to explain the results with the help of proposed hypotheses. The discussion is currently incomplete and inadequate.

**Reviewer #2:** Re: EEG Sonification Improves Sleep Staging Performance in Novice Stagers

General comments

Manuscript “EEG Sonification Improves Sleep Staging Performance in Novice Stagers” aimed at investigating whether adding an auditory representation (sonification) of the EEG to a standard visual representation could improve sleep staging performance. Additionally, the study examines if this method reduces workload. The authors found that adding sonification did not result in improvements in accuracy, speed, or workload for the entire subject group .However we found sonification improved accuracy for the least experienced participants. This study explores an interesting and promising concept. Below are some suggestions that should be considered by the authors of the manuscript:

Specific comments

Abstract

1. In the abstract, also mention the classification of groups based on the level of experience.

Methods

2. How many groups did you have in total? Please provide a detailed explanation of how the groups were divided based on the level of experience and the reception of the Sanifcation sound.

Introduction

1. The first paragraph of the “Manual Sleep Staging” section needs more references.

2. Hypothesis 1 should be written in full. In which aspect does it perform better?

Statistical analysis

1. Why is only the prior year experience of the participants considered? It seems that if the overall experience of the participants is taken into account, the accuracy of the study would be higher.

Discussion

Overall the discussion is not well written.

1. The consequences of the findings in relation to the hypotheses proposed in this study should be clarified with the existing literature.

2. How do you justify the effectiveness of Sanification for novices? You need to reference relevant articles on this.

3. Paragraph 5 is not relevant to this section. It is suggested that it be briefly mentioned in the Method section.

6. PLOS authors have the option to publish the peer review history of their article (what does this mean?). If published, this will include your full peer review and any attached files.

Reviewer #1: No

Reviewer #2: No

---

## [Author Response · Author response to Decision Letter 1]

19 Jul 2025

Thank you for your comments! We have improved the paper accordingly as described below.

Journal Requirements:

We have updated the paper to use the PLoS Latex template and formatting style

3. Thank you for uploading your study's underlying data set. Unfortunately, the repository you have noted in your Data Availability statement does not qualify as an acceptable data repository according to PLOS's standards.

We have uploaded the data to Dryad

Reviewer 1 comments

Reviewer #1: This study is an interesting study on the effect of auditory intervention on sleep staging that has not been investigated so far. Most studies in the field of sanitization have used this type of intervention to develop (cognitive-motor) performance and have shown positive results.

1- The abstract is well-written; however, it would be beneficial to include a brief one-line reference to the feature that was presented to participants as an auditory pattern (sonification).

We have added this to the abstract

2- The introduction is well-structured. However, it would be beneficial to examine the effects of sonification on other functions, including motor performance, before discussing research related to sleep staging performance. Numerous studies have been conducted in this regard, some of which I have referenced. For esample:

Ramezanzade, H. (2019). Adding Acoustical to visual movement patterns to retest whether imitation is goal- or pattern- directed. Perceptual and Motor Skills, 127(1), https://doi.org/10.1177/0031512519870418.

Parimi F, Abdoli B, Ramezanzade H, Aghdaei M (2024) The Effect of Internal and external imagery on learning badminton long serve skill: The role of visual and audiovisual imagery. PLoS ONE 19(9): e0309473. https://doi.org/10.1371/journal.pone.0309473.

Effenberg AO, Fehse U, Schmitz G, Krueger B, Mechling H. Movement sonification: effects on motor learning beyond rhythmic adjustments. Frontiers in neuroscience. 2016;10:219. pmid:27303255.

3- You can reference the following source for the statement "audition is more temporally precise, while vision is more spatially precise".

Ramezanzade, H., Abdoli, B., Farsi, A., & Sanjari, M. A. (2017). The effect of sonification of visual modeling on relative timing: The role in motor learning. Acta Kinsiologica, 11, Supp. 2, 17-27.

4- You can also reference the following source for the statement “This is also confirmed by experimental results which have demonstrated that sound can enhance visual perception”.

Ramezanzade, H., Abdoli, B., Farsi, A., & Sanjari, M. A. (2017). The effect of sonification modelling on perception and accuracy of performing jump shot basketball. International Journal of Sports Studies, 4(11), 1388-1392.

Effenberg AO. Movement sonification: Effects on perception and action. IEEE multimedia. 2005;12(2):53–9.

We have added the Parimi at al (2024), Effenberg et al (2016) and Effenberg et al (2005) references and added more discussion on the effects of sonification on other cognitive and motor tasks

5- Please specify precisely why the O1, F1, and C3 regions were selected for generating the auditory pattern.

These regions were used because they are the standard electrode locations used for sleep staging in the AASM system–we have added this to the paper.

6- The methodology is well-written, but it should include more details regarding the physical characteristics of the sound (frequency, volume, pitch, timbre, etc.)

We have added these details to the stimulus generation procedure. We have also provided examples of the generated files and corresponding visual EEG traces.

7- It would be better to describe sleep staging precisely and in detail in both the introduction and methodology sections, while also specifying the characteristics of each stage.

We have added more discussion of sleep staging and the characteristics of each stage to these sections

8- The results section is well-reported. The discussion is also good, but it would be better to provide more speculation regarding the reasons behind the obtained results. It would be beneficial to study recent research related to sleep and its influencing variables, and to explain the results with the help of proposed hypotheses. The discussion is currently incomplete and inadequate.

We have revised the discussion section to be clearer about our interpretation of the results–in particular, why we think we observed a benefit for inexperienced novice sleep stagers but not for experienced sleep stagers. However we don’t think it would be helpful to talk about the factors that influence sleep as the focus of this study is primarily on sleep staging as a perceptual learning task, not on sleep itself.

Reviewer 2 comments

Manuscript “EEG Sonification Improves Sleep Staging Performance in Novice Stagers” aimed at investigating whether adding an auditory representation (sonification) of the EEG to a standard visual representation could improve sleep staging performance. Additionally, the study examines if this method reduces workload. The authors found that adding sonification did not result in improvements in accuracy, speed, or workload for the entire subject group .However we found sonification improved accuracy for the least experienced participants. This study explores an interesting and promising concept. Below are some suggestions that should be considered by the authors of the manuscript:

Specific comments

Abstract

1. In the abstract, also mention the classification of groups based on the level of experience.

We have added this to the abstract

Methods

2. How many groups did you have in total? Please provide a detailed explanation of how the groups were divided based on the level of experience and the reception of the Sanifcation sound.

We defined five experience groups based on polysomnograms scored in the last year –these are the groups shown in table 1. We have updated the methods to clarify that these were the groups participants were divided into.

Introduction

1. The first paragraph of the “Manual Sleep Staging” section needs more references.

We have added more references and explanation of the process of sleep staging

2. Hypothesis 1 should be written in full. In which aspect does it perform better?

We have clarified that this is referring to increased agreement with the gold standard scorers as measured by cohen’s kappa

Statistical analysis

1. Why is only the prior year experience of the participants considered? It seems that if the overall experience of the participants is taken into account, the accuracy of the study would be higher.

We chose to measure the prior year as (1) we felt that people would be less precise when estimating the total number of nights staged across their career and (2) recent practice in sleep staging would be the most relevant. We have also added this to the paper.

Discussion

The consequences of the findings in relation to the hypotheses proposed in this study should be clarified with the existing literature.

We have further elaborated on how these results relate to other sonification studies and clarified our explanation of why we think we see results only for novice sleep stagers.

How do you justify the effectiveness of Sanification for novices? You need to reference relevant articles on this.

This is based purely on our observation that novices (people who had scored 10 polysomnograms or less) performed better with sonification than without sonification in our study. We have clarified this in the text.

Paragraph 5 is not relevant to this section. It is suggested that it be briefly mentioned in the Method section.

We have moved this paragraph

---

## [Editor Report · Decision Letter 1]

22 Jul 2025

EEG Sonification Improves Sleep Staging Performance in Novice Stagers

PONE-D-25-11016R1

Dear Dr. Whitmore,

We’re pleased to inform you that your manuscript has been judged scientifically suitable for publication and will be formally accepted for publication once it meets all outstanding technical requirements.

Kind regards,

Hesam Ramezanzade, Ph.D

Academic Editor

PLOS ONE
---

## [Editor Report · Acceptance letter]

PONE-D-25-11016R1

PLOS ONE

Dear Dr. Whitmore,

I'm pleased to inform you that your manuscript has been deemed suitable for publication in PLOS ONE. Congratulations! Your manuscript is now being handed over to our production team.

Kind regards,

on behalf of

Dr. Hesam Ramezanzade

Academic Editor

PLOS ONE